# Follicular Lymphoma Microenvironment Traits Associated with Event-Free Survival

**DOI:** 10.3390/ijms24129909

**Published:** 2023-06-08

**Authors:** Maria Maddalena Tumedei, Filippo Piccinini, Irene Azzali, Francesca Pirini, Sara Bravaccini, Serena De Matteis, Claudio Agostinelli, Gastone Castellani, Michele Zanoni, Michela Cortesi, Barbara Vergani, Biagio Eugenio Leone, Simona Righi, Anna Gazzola, Beatrice Casadei, Davide Gentilini, Luciano Calzari, Francesco Limarzi, Elena Sabattini, Andrea Pession, Marcella Tazzari, Clara Bertuzzi

**Affiliations:** 1Biosciences Laboratory, IRCCS Istituto Romagnolo per lo Studio dei Tumori (IRST) “Dino Amadori”, 47014 Meldola, Italy; 2Scientific Directorate, IRCCS Istituto Romagnolo per lo Studio dei Tumori (IRST) “Dino Amadori”, 47014 Meldola, Italy; 3Department of Medical and Surgical Sciences (DIMEC), University of Bologna, 40126 Bologna, Italy; 4Biostatistics and Clinical Trials Unit, IRCCS Istituto Romagnolo per lo Studio dei Tumori (IRST) “Dino Amadori”, 47014 Meldola, Italy; 5Immunobiology of Transplants and Advanced Cellular Therapies Unit, IRCCS Azienda Ospedaliero-Universitaria di Bologna, 40138 Bologna, Italy; 6Hematopathology Unit, IRCCS Azienda Ospedaliero-Universitaria di Bologna, 40138 Bologna, Italy; 7School of Medicine and Surgery, University of Milano-Bicocca, 20900 Monza, Italy; 8Hematology Unit, IRCCS Azienda Ospedaliero-Universitaria di Bologna, 40138 Bologna, Italy; 9Department of Brain and Behavioral Sciences, Università di Pavia, 27100 Pavia, Italy; 10Bioinformatics and Statistical Genomics Unit, Istituto Auxologico Italiano IRCCS, 20095 Cusano Milanino, Italy; 11Pathology Unit, Morgagni-Pierantoni Hospital, AUSL Romagna, Via Carlo Forlanini, 34, 47121 Forlì, Italy; 12Department of Pediatrics, IRCCS Azienda Ospedaliero-Universitaria di Bologna, 40138 Bologna, Italy; 13Immunotherapy Cell Therapy and Biobank (ITCB) Unit, IRCCS Istituto Romagnolo per lo Studio dei Tumori (IRST) “Dino Amadori”, 47014 Meldola, Italy

**Keywords:** Follicular Lymphoma, immunohistochemistry, tumor-associated macrophages, progressive disease, tumor microenvironment, digital pathology

## Abstract

The majority of patients with Follicular Lymphoma (FL) experience subsequent phases of remission and relapse, making the disease “virtually” incurable. To predict the outcome of FL patients at diagnosis, various clinical-based prognostic scores have been proposed; nonetheless, they continue to fail for a subset of patients. Gene expression profiling has highlighted the pivotal role of the tumor microenvironment (TME) in the FL prognosis; nevertheless, there is still a need to standardize the assessment of immune-infiltrating cells for the prognostic classification of patients with early or late progressing disease. We studied a retrospective cohort of 49 FL lymph node biopsies at the time of the initial diagnosis using pathologist-guided analysis on whole slide images, and we characterized the immune repertoire for both quantity and distribution (intrafollicular, IF and extrafollicular, EF) of cell subsets in relation to clinical outcome. We looked for the natural killer (CD56), T lymphocyte (CD8, CD4, PD1) and macrophage (CD68, CD163, MA4A4A)-associated markers. High CD163/CD8 EF ratios and high CD56/MS4A4A EF ratios, according to Kaplan–Meier estimates were linked with shorter EFS (event-free survival), with the former being the only one associated with POD24. In contrast to IF CD68+ cells, which represent a more homogeneous population, higher in non-progressing patients, EF CD68+ macrophages did not stratify according to survival. We also identify distinctive MS4A4A+CD163-macrophage populations with different prognostic weights. Enlarging the macrophage characterization and combining it with a lymphoid marker in the rituximab era, in our opinion, may enable prognostic stratification for low-/high-grade FL patients beyond POD24. These findings warrant validation across larger FL cohorts.

## 1. Introduction

Follicular Lymphoma (FL) represents the most common type of indolent non-Hodgkin lymphoma, accounting for about 7–15% of all lymphomas worldwide [1], with the highest incidence in the USA and Western Europe [2]. The two types of B cells that make up neoplastic follicles in FL are typically centrocytes and centroblasts [3], with the latter’s abundance serving as the basis for disease grading [4,5,6]. A t(14;18)(q32;q21) translocations that promote BCL2 gene expression is present in around 85% of cases. However, a small percentage of FL cases are BCL2 negative and exhibit other markers, such as CD10, BCL6, p53, Ki-67 and MUM-1 [7]. Even if FL patients can achieve long-term survival, the disease is marked by a “wax and wane’’ clinical pattern with high response rates hindered by frequent disease recurrence or progression to large B-cell lymphoma, which has a worse prognosis in about 35% of cases [1]. Many efforts have been made in order to predict the prognosis and risk of evolution in FL patients as a result of this clinical scenario. 

The FLIPI (Follicular Lymphoma international prognostic index) clinical score and its evolution to FLIPI2, also applicable to rituximab-treated cases, are widely used in clinical practice to categorize patients in risk groups (low, intermediate and high) [2,8]. These scores proved to be reliable and easy to apply, but they ignored the biological features of the disease and only took into consideration clinical parameters, such as age, Ann Arbor stage, hemoglobin level, number of nodal sites and serum lactate dehydrogenase (LDH) level [8]. In order to strengthen therapeutic strategies, further insights into the biology of FL are required as there is still outcome heterogeneity within each risk group. In 2015, Pastore and Colleagues suggested an integrated approach that includes both the evaluation of genetic alterations and clinical parameters [9]. They established the m7-FLIPI clinical–genetic risk model, which considers the mutation status of seven genes: *EZH2*, *ARID1A*, *MEF2B*, *EP300*, *FOXO1*, *CREBBP* and *CARD11*. Despite its effectiveness, this specific type of individual genetic evaluation is not practical in the clinicopathological setting [10]. The most effective unfavorable prognostic indicator available at the moment is the progression of the disease within 24 months (POD24) from the first-line treatment [11]. POD24, on the other hand, only captures events that occurred over a two-year period, leaving out all the other patients at risk of progression. The tumor microenvironment (TME) is crucial to the FL prognosis, as shown by gene expression profile (GEP)-based studies [6,12]. Tumor-associated macrophages (TAM) marked by CD68, CD163 [13,14], CD4+ and CD8+ T cells, non-neoplastic B cells and CD21+ follicular dendritic cells (FDC) [15] are the components of FL TME. Pure FL cell lines cannot be cultured in vitro without survival signals from feeder or cytokines, as it occurs in non-neoplastic germinal centers, where these host cells are essential for the development and maintenance of B-cell response [16]. Dave et al. described the TME impact on FL biology. They identified two different immune response (IR) signatures, that independently predicted the outcome of FL. A primary signature, specifically associated with long survival, included T cells together with macrophage and FDC markers, whereas a secondary signature was predominantly linked to short survival and involved genes encoding for TAMs [12]. In our work, we performed a comprehensive TME profiling, which was specifically characterized by a pathologist-guided digital spatial in situ analysis of FL samples at the time of first diagnosis (Appendix A). We then assessed its correlation with patient clinical and pathological characteristics, genetic FL features and clinical outcome.

## 2. Results

### 2.1. Clinical and Pathological Characteristics of Patients

A total of 49 FL patients (26 men and 23 women) with sufficient formalin-fixed paraffin-embedded (FFPE) material were included in the study. Overall, the clinical and pathological characteristics of the patients were well-balanced (Table 1 and Appendix A). The histological distribution according to the World Health Organization (WHO) classification was as follows: grade I–II, 24 (48.97%) and grade IIIA, 25 (51.03%) [5]. A total of 10 patients (20.40%) presented with stage I, 11 (22.45%) with stage II, 9 (18.37%) with stage III and 19 (38.78%) with stage IV. The FLIPI score at diagnosis was High (H), Intermediate (I), and Low (L) in 6 (12.25%), 22 (44.9%) and 21 (42.85%) patients, respectively. In line with the BCL2 translocation event, our cohort was divided into 13 (26.53%) BCL2 negative and 36 (73.47%) BCL2 positive patients. 

### 2.2. Abundance of CD68+ Macrophages in the Intrafollicular Area Correlates with Event-Free Survival

A pathologist-guided analysis of CD68 (PGM1 clone) marker expression was conducted on FL lymph node biopsies collected at the time of diagnosis. To this aim, standard IHC followed by whole-slide-image (WSI) acquisition was performed (Appendix A). CD68 positive expression was estimated both in the intrafollicular (IF) as well as in the extrafollicular (EF) areas (Figure 1 and Figure 2). According to WHO guidelines for centroblast enumeration within follicles for FL grading [4], the quantification of positive cells was retrieved from 10 randomly selected microscopic high-power fields (HPFs; 40× magnification). IF CD68+ cell count ranged from 12 to 67.5. Their frequencies were lower compared to EF CD68+ cells, accounting for 22 to 103.5 (Appendix A). IF and EF CD68 IHC values were evaluated according to relevant clinical and pathological variables, namely grade, stage, FLIPI, BCL2 and POD24 (Figure 1a–e and Figure 2a–e). Neither IF nor EF CD68+ cells were associated with grade (IF: *p*-value 0.94; Figure 1a, EF: *p*-value 0.96; Figure 2a), while only EF CD68+ cells exhibited significantly different values between stage II and III (EF: *p*-value 0.02; Figure 2b). When compared to the FLIPI High/Intermediate group, the subgroup of patients classified as FLIPI Low had evidence of a higher level of EF CD68+ (EF: *p*-value 0.04; Figure 2c). Of biological relevance, BCL2 negative cases display a significantly higher value of both IF and EF CD68+ cells rather than BCL2 positive cases (IF: *p*-value 0.003; EF: *p*-value 0.002; Figure 1d and Figure 2d). Furthermore, no significant difference was observed between POD24 positive and negative patients in both IF and EF areas (IF: *p*-value 0.48; EF: *p*-value 0.93, Figure 1e and Figure 2e). Conversely, considering a long-term follow-up endpoint, IF CD68+ cells abundance beyond the cutpoint 32.5/HPF showed a significantly longer EFS than lower levels (5-year EFS: 0.90, 95% CI 0.73–1.00 vs. 5-year EFS: 0.40, 95% CI 0.26–0.62; *p*-value 0.01; Figure 1f). On the other hand, no association with EFS was found for EF CD68+ cells (Figure 2f). Figure 1g and Figure 2g, respectively, show representative bright-field images of the IF and EF CD68 expression pattern for patients who either had or did not develop a progressive/recurrent disease (PD-positive and PD-negative patients). Sequential IHC (Appendix A and Materials and Methods) was applied including the CD21 FDC marker (gray) to further highlight the compartmentalization of these CD68+ macrophages (green) within the germinal center (Figure 1h). Representative pseudo-fluorescent images, at low (left) and high magnification (right), show the higher abundance of round shape CD68+ macrophages localized within the FDC meshworks in PD-negative patients. No evidence of CD21 and CD68 co-localization was found. 

### 2.3. CD163/CD8 EF Ratios Are Significantly Associated with EFS

Given the macrophage plasticity across cancers, we decided to further characterize the functional polarization of FL macrophages utilizing CD163 as a broadly applied marker of tumor-associated myeloid cells. Sequential IHC (Appendix A, Panel A) was used to combine CD163 (red) staining with the previously analyzed CD68 (green) and CD21 (gray) expression (Figure 2h). In addition to the previous findings, reporting higher EF CD68+ cells, we provided more evidence of the heterogeneity of this EF macrophage population encompassing more than one single phenotype. Indeed, pseudo-color images show the coexistence of CD163 (red) or CD68 (green) single-positive cells together with double-positive CD163+CD68+ cells (yellow), although IF CD68 cells appeared to be CD163-negative. Considering this different expression and relative spatial distribution we decided to deeply score this CD163+ macrophage population, potentially resembling pro-tumor FL myeloid cells. In keeping with the objectives of our research, we chose to create an automatic staining approach that is easily applicable in clinical settings. Given that CD163+ cells have been shown to reduce T-cell abundance [17,18,19], a double labeling procedure using CD163 (brown) and the anti-tumor effector T cell marker CD8 (red), was carried out on the entire patient cohort (Appendix A, Figure 3a). Despite the clinical–pathological features, IF areas were largely absent of CD163+ cells and CD8+ cells, which is consistent with the above data. Ten ROIs were used to score the cells at a 40× magnification, and the ratio of CD163 to CD8 cells was plotted. Representative bright-field IHC images of the EF double staining are illustrated in Figure 3a. Clinical variables and EF IHC values were evaluated for correlation. Notably, there was a significant association between POD24 and the CD163/CD8 EF ratio (*p*-value 0.03; Figure 3b). Moreover, EF CD163/CD8 ratios equal or greater than the cut-off 16.3/HPF were significantly associated with inferior EFS as compared to lower values (5-year EFS: 0.23, 95% CI 0.09–0.62 vs. 5-year EFS: 0.67, 95% CI 0.51–0.87; *p*-value < 0.001; Figure 3c). No other comparison was statistically significant (Figure 3d–g). Following the standardization of EF regions among ROIs/Patients, sequential IHC was used for an enhanced automatic quantification, and subsequently confirmation of the EF CD163/CD8 ratios (Figure 4a). Particularly, five CD21 original ROIs were used as masks for each patient during a pathologist-guided software-based segmentation of EF (white) areas (Figure 4b). Image cytometry was used to quantitatively assess CD163 (cyano) and CD8 (red) in specific EF regions to produce flow cytometry-like dot plots (Figure 4c). The pathologist score ratio obtained on the entire cohort was corroborated by the plotted data (Figure 4d) of the automatic analysis.

### 2.4. CD163-MS4A4A+ Macrophages Are Inversely Related to EFS

As we observed FL TAM diversity in the TME, we decided to stain our tissue with a second myeloid/lymphoid marker combination recently described across solid tumors [20]. Based on this literature, we tested the NK cell marker CD56 and the macrophage-specific tetraspan molecule, MS4A4A. MS4A4A (brown) and CD56 (red) were validated in automated double staining and applied to the whole patient cohort. The CD56/MS4A4A ratio was calculated using standard pathology procedures, dividing the lower population over the most abundant one. Once more, 10 EF areas at HPF magnification for each FFPE slide were taken into account for calculating the percentage of cell ratio. Figure 5a shows a representative bright-field IHC image of the EF double staining. Clinical variables and EF IHC values were evaluated for correlation. Of note, the analysis conducted according to clinical variables highlighted only a statistical significance in relation to the *BCL2* positivity (*p*-value 0.04; Figure 5b–e). Clinical outcomes showed a significant correlation between CD56/MS4A4A EF ratio equal to or greater than the cut-off 18/HPF and inferior EFS (5-year EFS: 0.10, 95% CI 0.02–0.64 vs. 5-year EFS: 0.66, 95% CI 0.49–0.88; *p*-value < 0.001; Figure 5g). However, no association was found with the POD24 event (Figure 5f). The findings demonstrated a correlation between high CD163/CD8 and CD56/MS4A4A IHC ratio values and a lower EFS endpoint. We performed sequential IHC (Appendix A, Panel B) to thoroughly explore this result, given also the differential numerical position within the computed ratio, and to understand if this discrepancy was truly caused by the macrophage markers designating for different FL EF TAM. We stained PD-negative cases expecting to display many MS4A4A+ cells. Indeed, pseudo-color images demonstrated the coexistence of CD163+MS4A4A+ cells (magenta), but also the presence of MS4A4A single-positive cells (blue) (Figure 5h). No co-localization of CD21 and MS4A4A expression was observed. 

### 2.5. IF CD4+PD-1hi T Cells Are Associated with Longer EFS

In line with previous works, we also assessed the relative expression of intratumoral PD1+ cells (Figure 6) [21,22]. Bright-field IHC, evidenced the PD1 expression in both IF and EF areas (Figure 6a). IF PD1’s immunoreactivity was higher than EF PD1’s signal (Figure 6a). No significant association of IF and EF PD1 positivity was observed in relation to all the analyzed clinical variables, including the POD24 event (Figure 6b,c). Only IF PD1 was significantly associated with EFS; in particular, IF PD1 equal or greater than the cut-off 53.75/HPF was associated with better EFS as compared to lower values (5-year EFS: 0.77, 95% CI 0.59–0.995 vs. 5-year EFS: 0.35 95% CI 0.19–0.64; *p*-value < 0.03; Figure 6d). No association was found for EF PD1 scores (Figure 6e). In accordance with the notion of IF follicular helper T (Tfh) cells already described by others [22], multiplex IHC confirmed the nature of IF PD-1 high T cells as CD4+ cells, (Figure 6f and Appendix A, Panel C). Even though our data highlight the expression of two distinct PD1 positive intra-tumoral populations (PD1 high vs. PD1dim), each of which displayed a distinctive pattern of in situ distribution, we were unable to confirm previous research reporting the association of EF PD1dim T cells with inferior outcome [21]. Furthermore, in contrast with the finding that Tfh cells may support the survival of lymphoma B cells [21], high numbers of IF CD4+PD1hi cells were associated with prolonged EFS in our analysis, which is consistent with another work by Smeltzer and Colleagues [23].

### 2.6. Evaluation of FL Microenvironment Traits Effect on EFS

Univariate Cox regression models were used to obtain an estimate of each marker effect on EFS and to identify potential baseline clinical confounders (Table 2). CD163/CD8 EF ratio levels lower than 16.3/HPF were associated with better EFS (HR 0.23, 95% CI 0.09–0.57, *p*-value = 0.002). Analogous results were recorded for CD56/MS4A4A EF ratio, where values lower than 18/HPF were associated with better EFS (HR 0.23, 95% CI 0.09–0.59, *p*-value = 0.002). With regard to CD68 abundances, CD68 IF lower values were associated with worse EFS (HR 8.72, 95% CI 1.17–64.82, *p*-value = 0.03), while no effect was detected for CD68 EF (HR 2.10, 95% CI 0.88–4.96, *p*-value = 0.10). No baseline clinical variables showed evidence of association with EFS. Multivariate analysis based on markers that were significantly correlated with EFS did not reveal any independent effect (Table 2). 

## 3. Discussion

The natural history of FL is characterized by a pattern of frequent relapses with a variable response duration and an increased likelihood of shorter survival after each relapse as well as an evolution towards a more aggressive disease such as large B cell Lymphoma (DLBCL) [5]. Italy accounts for 3000 new diagnoses of FL each year, and the incidence of this hematologic neoplasm is currently increasing [24]. Different prognostic scores have been proposed to predict the outcome of FL patients at diagnosis. Nowadays, only clinical-based indexes, such as FLIPI, FLIPI2 or POD24 are used [2,8,11,25]. Even though hematologists rely on these relatively robust clinical scores, there is still extreme heterogeneity in the clinical course within each risk class.

Furthermore, POD24, even if an association with an inferior OS has been validated, requires two years’ evaluation and the administration of a first unsuccessful line of therapy [26,27]. Influence by non-neoplastic elements on FL prognosis has already emerged [7,28,29]. Indeed, immune cell subsets have been described to both promote immunosuppression and help in tumor cell clearance [6]. However, a comprehensive evaluation of FL immunobiology to anticipate patients experiencing progression or recurrence still lacks clinicopathological applicability. The aim of this study was to explore the FL TME to pinpoint those markers, or combinations thereof, informative to the routine diagnostic practice and quick to apply in the setting of therapeutic strategy for newly diagnosed patients. We focused on counting and correlating various immune cell subsets strictly with respect to their distribution as well as addressing their relevance in connection to clinical variables and outcomes (Table 1 and Appendix A). To better outline immune subtype identity, our work specifically examined both macrophage and lymphoid markers implementing bright-field IHC with iterative immunostaining for various markers on the same FFPE slide. Whole slide image analysis was preferred to the often applied tissue microarray (TMA) core analysis which may result in a biased evaluation [2,28,29,30,31], given the FL germinal center organization in three-dimensional structures (follicles). Moreover, and in contrast to prior research [11], we looked at spatial immune infiltration in relation to the POD24 event as well as a longer follow-up to uncover the association with risk progression or recurrence also after 24 months. In most cases, it has been observed that the presence of macrophages in a therapeutic regimen devoid of monoclonal antibodies has an unfavorable impact on patient prognosis; however, this effect is reversed by anti-CD20 antibody therapy. Nonetheless, in the post-rituximab era, different studies lead to variable results [1,7,16,32,33].

In our study, the prognostic effect of CD68+CD163- cells restricted to the CD21+ FDC distinctive meshwork staining pattern was only emphasized by the compartmentalized analysis of IF vs. EF areas. Our evaluation was carried out on the samples collected at the diagnosis and, even if most of the patients in our cohort (43/49) received rituximab as a first-line treatment, it emerged that the macrophage component is the one that has an impact on the patient’s prognosis. It was beyond the scope of this study to define whether the presence of rituximab in the therapeutic regimen counteracts a possible negative effect of the CD68+ subset within the germinal center or whether it performs a synergistic action with it in the elimination of the pathological elements. Nevertheless, taken together, our data underscore the relationship between IF CD68+ cells in the FL microenvironment and clinical outcome as opposed to those localized in EF areas, which did not appear to influence the disease course.

Further characterizing the myeloid component, we found that EF areas were enriched with FL-associated macrophages expressing CD163 and/or MS4A4A markers. Moreover, in line with previous reports [20], these markers delineated more than two separated macrophage subsets, not always co-expressing CD68. While CD163 has been already applied in FL, the same is not true for MS4A4A [13,28,30]. We chose to count these two major EF macrophage subsets in relation to the quantity of cytotoxic CD8+ T cells and CD56+ NK cells, respectively, for CD163- and MS4A4A-positive cells. Particularly, our data highlight the in situ imbalance of CD163+ cells compared to CD8+ T cells. Our proposed CD163/CD8 score is in line with and further enriches the already reported CD8 T cell and the CD163 scores described by others [1,13,31]. The multiplex IHC validates the inverse relation of CD163+ and MS4A4A+ cells, as mapping distinct intratumoral cell subsets, which is novel to the FL field, but underlines the prominent role of macrophages in this disease setting. The role of MS4A4A cells with respect to NK cells in FL needs to be addressed [20]. While the CD163/CD8 ratio speaks in favor of a supportive role of this marker combination in addressing the M2-like protumoral macrophages able to dampen CD8 T cell abundance and function. An explanation could be related to the antibody-dependent cell cytotoxicity (ADCC) role exploited by both NK and macrophage subsets upon monoclonal antibody therapy. Collectively, our multiplex IHC analyses highlight the existence of a MS4A4A+ distinct population that is not always limited to CD163+ macrophages and may have a different biologic function. Despite the fact that macrophages are known to be significant in FL, an implemented assessment of in situ FL macrophages will be crucial to determining the clinical relevance of different subpopulations more thoroughly. Indeed, when analyzing the IHC scores of macrophage subsets in relation to BCL2 expression the differential biologic role of these myeloid subpopulations is well evidenced. Although there was no link with CD163, it is worth noting that BCL2 negative cases had an enrichment of IF CD68+ cells as well as lower CD56/MS4A4A EF ratio in the same clinicopathological subgroup with higher values of both IF CD68+ and EF MS4A4A+ cells having a favorable prognosis (Figure 5e). The different pathogenesis of BCL2+ and BCL2− FL by itself may have an impact on the composition of the TME, even though the univariate analysis did not find an independent impact of BCL2 expression on EFS (Table 2). Because the *p*-value (0.06) is close to statistical significance, we assume that this is the case. Moreover, IHC examination revealed that CD163 immunoreactivity was mainly absent within the follicles, although MS4A4A+ cells were present, demonstrating the capacity of MS4A4A+CD68+ macrophages to infiltrate the follicles and influence the course of the disease.

It has to be acknowledged that the genome-wide approach gave us many insights into FL microenvironment biology [11,12]. Dave and colleagues identified two distinct signatures enriched in T-cell or macrophage genes and associated with longer or shorter survival, respectively. However, no evidence of lineage-specific macrophage marker genes, such as CD68, CD163 and MS4A4A in relation to PFS, was found. Similarly, our explorative molecular analysis conducted on a small cohort of patients (n = 18), for whom RNA was available from FL biopsies, did not underscore any T-cell- or macrophage-related genes associated with EFS or other clinical variables (Appendix A). Conversely, the nanostring analysis showed a reduction in the cytotoxic CD8+ T cell signature in PD-positive patients, consistent with the IHC-documented EF abundance of M2-like protumoral CD163+ macrophages (Appendix A). In contrast, IF PD1, equal or greater than the cut-off value of 53.75/HPF, was highlighted by our IHC analysis as being related to superior EFS, whereas EF PD1dim, which was expected to primarily indicate CD4+ and CD8+ exhausted T cells, was not significantly associated with EFS. It is worth noting that the same cells are known to express other markers, such as TIM3, which might be required to better outline exhaustion [21,23]. The impact of PD1+ elements (either with bright or dim expression) in FL is still widely debated [21,23]. Our data may be related to the patient cohort, which in our study included an equal distribution between a low and high grade and lack FL transformation towards a clinically aggressive lymphoma (e.g., DLBCL). T-cell exhaustion seems to be more responsible for turning off microenvironment’s suppressing effect on FL transformation, and thus, manifest more intensively in high-grade/diffuse FL [23]. Additional research is required to further understand the meaning of our findings. 

In addition to all of these microenvironmental factors, FL is an “epigenetic” disease characterized by mutations in chromatin regulator genes and epigenetic make-up can impact the TME composition [34]. To this aim, albeit for only a minor subset (n = 25) of patients (Appendix A, a multistep pipeline was used to detect methylation differences. A differential methylation analysis comparing PD-positive (PD_Y n = 9) vs. PD-negative (PD_N n = 16) failed to identify significantly deregulated CpG sites and regions (genes, promoters and CpG Islands) (Appendix A). Then, the burden of stochastic epimutations (SEMs) was evaluated; the number of SEMs in the PD positive group was slight, but significantly higher (generalized linear regression (glm) unadjusted model *p* = 0.0469) (Appendix A). To identify significantly SEM-enriched regions, a sliding window approach based on a hypergeometric test was carried out. After annotation of genomic positions, we obtained 276 SEM-enriched genes in the PD-positive group, and 591 gene loci for the PD-negative group. Appendix A, displays genes univocally belonging to the two groups (PD_Y, n = 165) and (PD_N, n = 480) (Appendix A). However, only two genes, DNMT3A and GPRC5B, were discovered to be FL-related and enriched in the PD_N group. However, none of the genes belongs to lineage-specific immune cell subsets or with the already identified immune signatures [12]. This suggests that despite the crucial role of genomic/epigenetic alterations of lymphoma cells in the development of the disease, the degree of immune infiltration is capturing aspects of FL biology that are distinct from its mutation and epigenetic profile. However, our epigenome-wide association study (EWAS) must be interpreted with caution, given the relatively small cohort of samples and the usage of FFPE, ultimately affecting data quality. Overall, our analysis underlines the importance of IHC spatial distribution of immune cell subsets not retrievable by gene expression study.

In summary, our analyses showed that FL immune subsets inside and outside the follicles have a diverse impact on EFS. This emphasizes how crucial it is to separately analyze IF and EF areas. Certainly, digital spatial profiling techniques, in use at the research level, might fill this gap, but are not suitable in terms of time and cost to be incorporated into the diagnostic laboratories and for the frontline management of patients. We believe that our proposed IHC subset analysis, including CD163 MS4A4A and CD68 for macrophage and measuring their level with respect to specific lymphoid markers, can aid in the prognostic stratification. As previously mentioned, all patients in our cohort except for six received rituximab-based systemic therapy as a first-line treatment. Validation of our results was also separately conducted performing a sensitivity analysis to assess the potential impact of surgery-treated patients on marker’s effect. The observed differences on EFS according to CD68 IF, CD163/CD8 EF, CD56/MS4A4A EF and PD1 IF values remained robust to the exclusion of the only surgery-treated/wait-and-see patients (Appendix A). This work was not powered to detect an effect of interest; thus, all the results must be interpreted as hypothesis-generating and the investigated associations must be assessed in a future study with a formal sample size determination. Future studies should also be focused on the identification of a marker prognostic signature across low-grade FL and beyond POD24; supporting the potential robustness of tools based on simple and cost-effective investigations that can be performed in a reproducible manner and directly provide clinicians precious prognostic information alongside the diagnosis of the disease. 

## 4. Materials and Methods

### 4.1. Study Population

The study was approved by the Local Ethical Committees of Area Vasta Romagna (CE 194/2021/Oss/AOUBo) and “L. e A. Seràgnoli” (Bologna) and informed consent was obtained from each patient for their biological material to be used for research purposes. FL classification, including histopathology and immunophenotyping, was performed at the Institute of Hematology “L. e A. Seràgnoli” according to the World Health Organization classification [4]. The median age at diagnosis was 58 (34–87) years; 26 (53.06%) patients were male and 23 (46.94%) were female. The exclusion criteria were: (1) aged under 18 years, (2) immunodeficient patients, (3) previous diagnosis of Lymphoma other than FL, (4) previous chemotherapy for any kind of neoplasms, (5) HIV, HBV or HCV seropositivity (Table 1 and Appendix A). A total of 51 patients were screened for inclusion and 49 FFPE blocks collected at the time of diagnosis were finally considered for the retrospective study. IHC analyses were performed on 49/49 (the number of assessed cases for each marker/ratio is specified in the Results Section). Pharmacological treatment after initial surgical resection was applied in all but three patients and depended on local institutional regimens. This included chemotherapy such as cyclophosphamide, doxorubicin, vincristine and prednisone (CHOP), anti-CD20 therapy (rituximab) or other treatments.

### 4.2. Automated Immunohistochemistry (IHC)

Tissue samples obtained through surgery were fixed in neutral buffered formalin and embedded in paraffin. Four-micron sections were mounted on positive-charged slides (Bio Optica, Milan, Italy). Immunostaining for biomarkers was performed at the Biosciences Laboratory of IRST IRCCS (Meldola, Italy) using the Ventana Benchmark ULTRA staining system (Ventana Medical Systems, Tucson, AZ, USA) with the Optiview DAB Detection kit (Ventana Medical Systems). All used antibodies are listed in Appendix A. The Optiview DAB Detection kit (Ventana Medical Systems) and the UltraView Universal Alkaline Phosphatase Red Detection kit (Ventana Medical Systems) were used for single and double immunostaining. All the tissue sections were counterstained for 16 min with Hematoxylin II (Ventana Medical System) after each immunostaining. High-resolution WSI (40× magnification) of IHC stained slides were acquired using the Aperio CS2 slide scanner (Leica Biosystems Nussloch GmbH). Quantification of IHC stains was performed by two expert pathologists (C.B. and C.A.), and further validated by automated software-based tools. Myeloid and lymphoid populations were scored on the whole slide images selecting 10 high power fields (HPF) referring to the number of areas considered reliable for FL grading [4]. For single-marker quantification, as well as for ratio calculation, the number reported in graphs and tables (see raw data) refers to the absolute number/hpf. Quantifications of markers/ratio (reporting the less abundant marker as numerator) were performed considering IF and EF areas. 

### 4.3. Sequential Immunohistochemistry (seqIHC)

For sequential IHC, a non-biotin poly HRP conjugate system followed by aminoethyl carbazole (AEC) substrate reaction was used instead of 3,3-diaminobenzidine (DAB). All the AEC conditions were validated on tissue microarrays (TMAs) containing different positive control tissues or on human tonsils (Appendix A). Each marker was approved by an expert pathologist (F.L.) upon comparison with the DAB staining, performed on consecutive slides. For the iterative staining, a chromogen destaining step (in alcohol) and a stripping step (in citrate buffer) were applied according to a previously published protocol [35]. The best sequence (rounds) for each marker was determined (Appendix A). Whole slide images were acquired from each staining (Rounds) and Regions of Interest (ROIs) were selected for the computational image processing (Appendix A). Each selected ROI was manually analyzed by an expert pathologist (C.B.), which using the ImageJ ROI Manager plugin [36] segmented the EF and IF areas to prove that approximately they cover 50% of the selected ROIs. Figure 4a shows an example of segmented areas (the EF areas in white; the IF areas in black). Then, quantification and pseudo-color image visualization were conducted using in sequence 4 freely available tools: (a) DS4H Image Alignment, for aligning multimodal images and proceeding in the colocalization analysis [35]; (b) bUnwarpJ, for correcting the aligned images for 2D elastic deformations due to sample washing and staining [37]; (c) Colour Deconvolution, for stain un mixing [38]; (d) CellProfiler, for segmenting cells and extracting features [39]. Then, image cytometry data were generated using .csv files and the commercial FCS Express 7 tool (DeNovo Software) (Appendix A).

### 4.4. Statistical Analysis 

For continuous variables, the data were summarized by median and range and for categorical variables by frequency and percentage. Statistical analysis was performed using GraphPad Prism (version 9, Jolla, CA, USA). POD24, BCL2, FLIPI and grade were the clinical and biological parameters utilized to compare differences in the number of positive cells for a given marker or ratio using the Mann–Whitney U. Regarding Stage, Kruskall-Wallis test was used to evaluate the association with each given marker or ratio and Dunn’s multiple comparison test was used for the post hoc test. The Kaplan–Meier method was used to estimate EFS defined as the time from diagnosis to first documented progression or relapse of the disease. A log-rank test was used for curve comparisons. Cut points were determined using a recursive partitioning algorithm to stratify each marker or ratio in relation to EFS. The Cox proportional hazard regression model was used to quantify the association between variables and EFS. Multivariate analysis was performed with purely explorative intent to assess the independent effect of each of the variables associated with EFS in univariate analysis. The results are reported as HR and 95% CI. Overall, a two-sided *p*-value < 0.05 was considered statistically significant. Statistical analyses were performed using GraphPad Prism (version 9, Jolla, CA, USA) and R version 4.2.0.

## Figures and Tables

**Figure 1 ijms-24-09909-f001:**
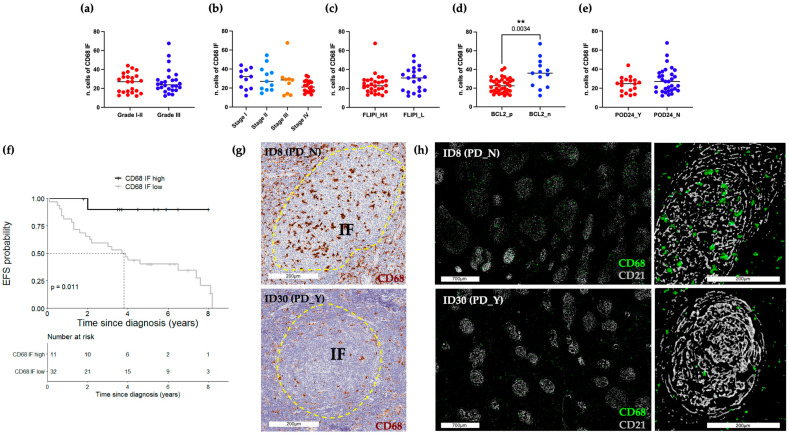
Abundance of IF CD68+ cells correlates with EFS. (**a**–**e**) BoxPlots of IF CD68 IHC scores. Each dot represents the median value of 10 regions of interest (ROIs) for each single patient. (**f**) EFS according to IF CD68+ stratified as high (n = 11) and low (n = 32) using the cut-off determined by recursive partitioning (32.5/HPF). Only values statistically significant are reported: ** *p* < 0.01 (**g**) Representative bright-field IHC images of CD68 immunoreactive cells of a PD_N (#ID8) and PD_Y (#ID30) patient. Yellow dotted lines delineate the follicle areas. Scale bars 200 μm. (**h**) Sequential IHC images showing high (#ID8) and low (#ID30) presence of CD68+ macrophages (green) within CD21 (gray) decorated follicles. High (scale bars 700 μm) and low (scale bars 200 μm) magnifications are reported. FLIPI, Follicular Lymphoma international prognostic index; H/I, High/Intermediate; L, Low; POD24, progression of disease within 24 months; EFS, event-free survival; PD, progressive/recurrent disease; IF, intrafollicular area.

**Figure 2 ijms-24-09909-f002:**
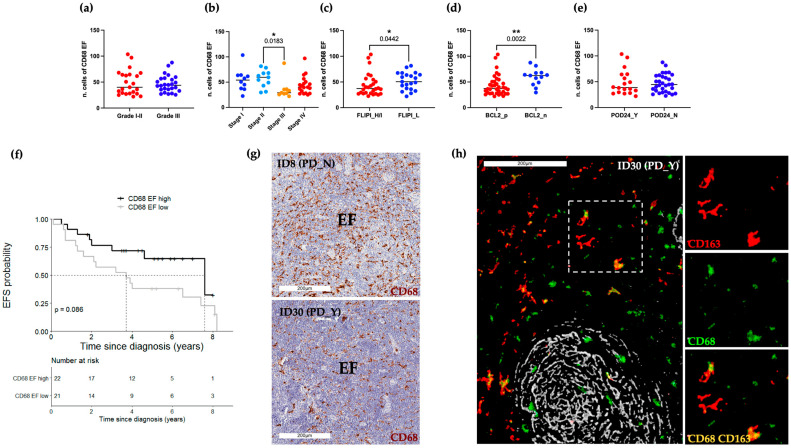
Frequency and characterization of EF CD68+ cells. (**a**–**e**) BoxPlots of EF CD68 IHC scores, where each dot represents the median value of 10 ROIs for each single patient; (**f**) EFS according to EF CD68+ stratified as high (n = 22) and low (n = 21) using the cut-off determined by recursive partitioning (39.75/HPF). (**g**) Representative bright-field IHC images of CD68 immunoreactive cells of a representative PD_N (#ID8) and PD_Y (#ID30) patient. Scale bars 200 μm. Only values statistically significant are reported: * *p* < 0.05, ** *p* < 0.01 (**h**) Pseudo-color images derived from sequential IHC show three distinct intratumoral macrophage populations: CD68 (green), CD163 (red) and double-positive CD163+CD68+ cells (yellow). FDCs decorated with the CD21 (gray) immunostaining highlight the IF vs. EF area. Scale bar 200 μm. High magnification inserts on the right show single-marker expression and the relative color merge. FLIPI, Follicular Lymphoma international prognostic index; H/I, High/Intermediate; L, Low; POD24, progression of disease within 24 months; EFS, event-free survival; PD, progressive/recurrent disease; EF, extrafollicular area.

**Figure 3 ijms-24-09909-f003:**
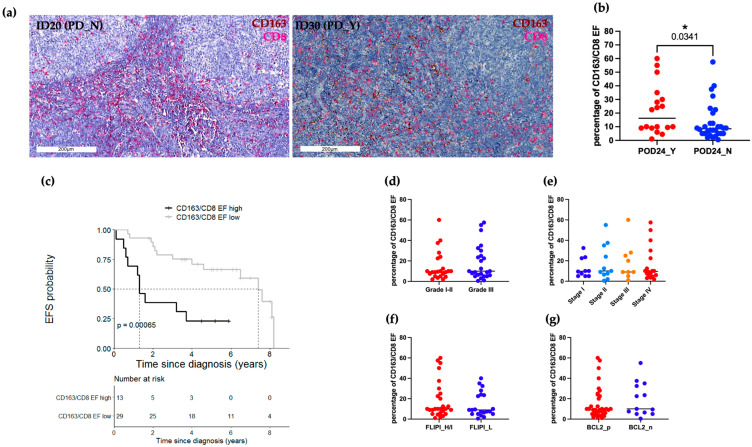
CD163/CD8 EF ratio correlates with EFS and significantly associates with POD24 event. (**a**) Representative bright-field IHC images of automated double IHC CD163 (brown) and CD8 (red) staining. Scale bars 200 μm. (**b**) Box plots of the percentage of CD163/CD8 EF ratio in relation to POD24 event, each dot represents the median value of 10 ROIs for a single patient. (**c**) High CD163/CD8 EF ratio (≥16.25/HPF; n = 13) showed a worse EFS than those with low values (n = 29). (**d**–**g**) Box plots of percentage of CD163/CD8 EF ratio, each dot represents the median value of 10 ROIs for each patient. PD, progressive/recurrent disease; POD24, progression of disease within 24 months; EFS, event-free survival; FLIPI, Follicular Lymphoma international prognostic index; H/I, High/Intermediate; L, Low. Only values statistically significant are reported: * *p* < 0.05.

**Figure 4 ijms-24-09909-f004:**
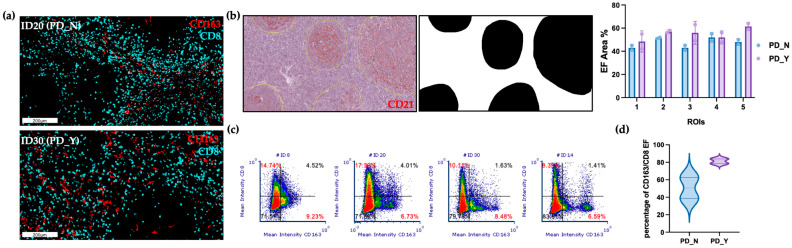
Software-based analyses of CD163/CD8 EF ratio. (**a**) Sequential IHC pseudo-color images of CD163 (cyano) and CD8 (red) cells within EF areas. Scale bars 200 μm. (**b**) Representative bright-field IHC CD21 ROI image used to create black and white masks for EF area segmentation and the segmentation results are indicated as a scatter dot plot of the percentage of EF occupied area across 5 ROIs for the four selected patients (PD_N: #ID8, #ID20; PD_Y: #ID30, #ID14). Statistical analysis was performed using 2-way ANOVA with a Šídák’s multiple comparisons test. (**c**) For each patient, an image cytometry-based cell population analysis for the lymphoid (CD8) and myeloid (CD163) biomarkers was obtained using the 5 ROIs previously described. Percentage of the CD8+ and CD163+ population are highlighted in red. (**d**) Violin Plot of percentage of CD163/CD8 EF ratio between 10 PD_N ROIs and 10 PD_Y ROIs. PD, progressive/recurrent disease; ROIs, regions of interest.

**Figure 5 ijms-24-09909-f005:**
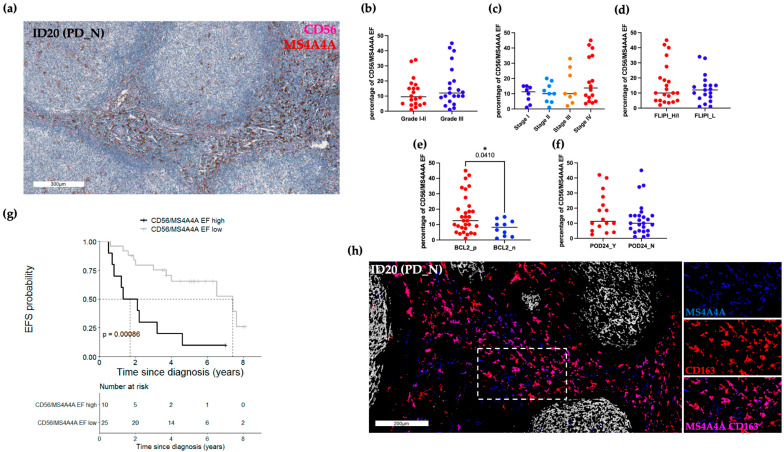
Low CD56/MS4A4A EF ratios are associated with longer EFS. (**a**) Representative automated double CD56 (red) and MS4A4A (brown) bright-field IHC image. Scale bar 300 μm. (**b**–**f**) Box plots depicting the percentage of CD56/MS4A4A EF ratios, each dot represents the median value of 10 ROIs for every patient. Only values statistically significant are reported: * *p* < 0.05 (**g**) High CD56/MS4A4A EF ratios (≥18/HPF; n = 10) showed a worse EFS than those with lower values (n = 25). (**h**) Representative sequential IHC pseudo-color images show three distinct intratumoral macrophage populations: MS4A4A (blue), CD163 (red) and double-positive MS4A4A+CD163+ cells (magenta). Scale bar 200 μm. High magnification inserts on the right display single-marker expression and the relative color merge. PD, progressive disease/recurrent; FLIPI, Follicular Lymphoma international prognostic index; H/I, High/Intermediate; L, Low; POD24, progression of disease within 24 months; EFS, event-free survival.

**Figure 6 ijms-24-09909-f006:**
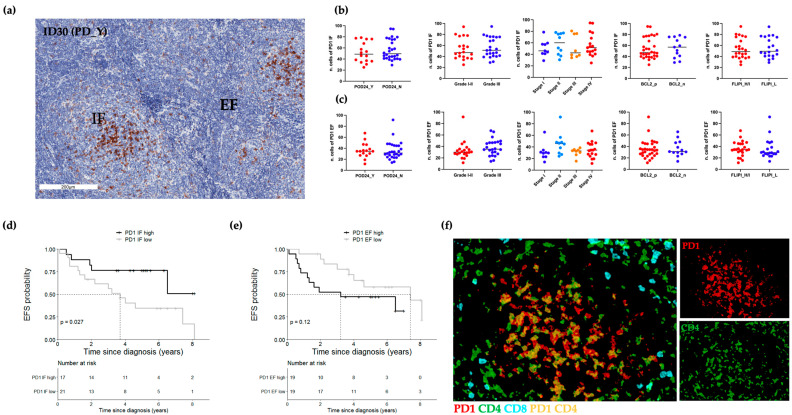
In situ PD1 expression in IF and EF regions. (**a**) Representative bright-field IHC image of PD1 in the IF and EF areas of a DR_Y case (#ID30). (**b**) Box plots of IF IHC scores, each dot represents the median value of 10 ROIs for each patient. (**c**) Box plots of EF IHC scores, each dot represents the median value of 10 ROIs for each patient. (**d**) High IF PD1 values (≥53.75/HPF; n = 17) showed a better EFS than those with lower values (n = 21). (**e**) Kaplan–Meier curves comparing EFS between PD1 EF high (≥34/HPF; n = 19) and PD1 EF low (n = 19). (**f**) Representative pseudo-fluorescence image of three-color staining: PD1 (red), CD4 (green) and CD8 (cyan) (#ID30). PD1+ cells were mainly IF CD4+ T cells (yellow). POD24, progression of disease within 24 months; EFS, event-free survival; PD, progressive/recurrent disease; FLIPI, Follicular Lymphoma international prognostic index; H/I, High/Intermediate; L, Low. Scale bars, 200 μm.

**Table 1 ijms-24-09909-t001:** Baseline patients’ characteristics.

Variable	Number (%)
Median age (years)	58 (34–87)
Gender	
Male	26 (53.06%)
Female	23 (46.94%)
Stage	
I	10 (20.40%)
II	11 (22.45%)
III	9 (18.37%)
IV	19 (38.78%)
FLIPI	
High (H)	6 (12.25%)
Intermediate (I)	22 (44.9%)
Low (L)	21(42.85%)
Grade	
I/II	24 (48.97%)
IIIA	25 (51.03%)
*BCL2*	
Negative	13 (26.53%)
Positive	36 (73.47%)

FLIPI, Follicular Lymphoma international prognostic index.

**Table 2 ijms-24-09909-t002:** Univariate and multivariate Cox regression analyses.

	Univariate			Multivariate		
	HR	95% CI	*p*-Value	HR	95% CI	*p*-Value
CD163/CD8 EF						
High ≥ 16.25/HPF	1.00			1.00		
Low < 16.25/HPF	0.23	0.09–0.57	0.002	0.38	0.13–1.09	0.07
CD56/M4A4A EF						
High ≥ 18/HPF	1.00			1.00		
Low < 18/HPF	0.23	0.09–0.59	0.002	0.51	0.16–1.60	0.25
CD68 IF						
High ≥ 32.5/HPF	1.00			1.00		
Low < 32.5/HPF	8.72	1.17–64.82	0.03	6.05	0.76–48.3	0.09
CD68 EF						
High ≥ 39.75/HPF	1.00					
Low < 39.75/HPF	2.10	0.88–4.96	0.10			
PD1 IF						
High ≥ 53.75/HPF	1.00			1.00		
Low < 53.75/HPF	2.97	1.08–8.20	0.04	1.52	0.44–5.20	0.51
PD1 EF						
High ≥ 34/HPF	1.00					
Low < 34/HPF	0.48	0.18–1.23	0.13			
Stage						
IV	1.00					
III	0.96	0.37–2.46	0.93			
II	0.28	0.06–1.25	0.09			
I	0.36	0.10–1.30	0.12			
Grade						
I/II	1.00					
III	1.37	0.60–3.11	0.45			
FLIPI						
H/I	1.00					
L	0.53	0.21–1.37	0.19			
BCL2						
*p*	1.00					
n	0.24	0.06–1.04	0.06			

FLIPI, Follicular Lymphoma international prognostic index; HR, hazard ratio; CI, confidence interval; HPF, high-power field; EF, extrafollicular; IF, intrafollicular.

## Data Availability

All raw data to the main and Appendix A are available upon request. Upon publication and if required all acquired images will be provided in anonymous format at: https://figshare.com/ and https://doi.org/10.6084/m9.figshare.c.6683513.v1.

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
