# Peer review of "Follicular Lymphoma Microenvironment Traits Associated with Event-Free Survival"

_ijms, 2023, doi:10.3390/ijms24129909_

Round 1
Reviewer 1 Report
Nice paper. It should thus be published
Reviewer 2 Report
Comments for Tumedei et al
The paper by Tumedei et al titled “Follicular lymphoma microenvironment traits associated with Event-Free-Survival” explores the follicular lymphoma (FL) tumor microenvironment (TME) to identify markers which as not yet been used routinely in the clinic for newly diagnosed FL patients.
Major and minor comments are denoted below.
Major Comments:
1. The abstract needs further clarification/rewriting since it sets the stage for the manuscript. The current version does not summarize clearly the points needs to be conveyed in this paper. Please note the following, they only representative examples:
a. Line 46 to 48 “EF CD68+ cell outline a more homogenous population, higher in non-progressing patients” This is a fragmented phrase, and it is not a complete sentence.
b. What is EF? Is it effector ratio? With that being said, please indicate in full words the abbreviations such as EF and IF.
c. If I am correct, CD163 is a scavenger receptor on macrophages and CD56 is a NK cell marker please indicate in the abstract.
2. The concept of tumor associated macrophages (TAMs) on FL transformation has long been established. Higher degree of intrafollicular (IF) infiltration of CD68+ and PD-L+ macrophages have been associated with a shorter time to FL transformation (De Palma et al., Cancer Cell 2013 and Mantovani et al., Nat Rev Clin Oncol. 2017). Moreover, CD163+ macrophages have been associated with a shorter PFS (Kridel et al Clin Cancer Res 2015).
Please explain the novelty of this manuscript.
3. EFS, which is defined as the time from randomization until the progression of diseases. Please indicate any data further clarifying cases which eventually progress.
4. Sample size too small to make conclusions
5. Need tSNE not only to justify the statements in lines 405 and 406, it is also needed to confirm all data scored by the pathologist since there could be biases.
Minor Comments:
6. Figure 1(a to c) please indicate p values. This also applies to all figures graphed in dot plots.
7. Please move Table 2 to supplemental.
8. Please indicate in larger fonts the magnification fields on all figures related to staining and immunofluorescence.
Reviewer 3 Report
This is a pilot study of 49 follicular NHL using diagnostic, pre-treatment biopsies. The authors applied an automated tissue- based phenotype spatial-oriented immune profiling. A shorter ESF appears to be associated with a high CD163/CDC8 ratio and a high CD56/MS4A4A ratio. The manuscript is carefully performed and is an interesting contribution in the field of follicular lymphoma.
Even for an experienced haemato-pathologist and clinical haematologist it is hard to rapidly understand the methodology. It’s like the forest is hidden behind too much trees.
My criticsms:
1) Pastore et al (lancet Oncology 2015) is named (line 79) but I don’t find the reference.
2) In M&M the authors have to show in a figure a workflow diagram with chronologic order of the different steps.
3) A schematic figure of bright-field IHC and iterative immune-staining the same FFPE slide has to be included.
Having performed these changes the paper’s interest increases for the IJMS readers.
Without that it is suitable for a pure pathology journal.
Round 2
Reviewer 2 Report
The authors have sufficiently addressed the reviewers comments. Therefore, I support the publication of this manuscript.